# Progesterone, Myo-Inositol, Dopamine and Prolactin Present in Follicular Fluid Have Differential Effects on Sperm Motility Subpopulations

**DOI:** 10.3390/life11111250

**Published:** 2021-11-17

**Authors:** Shannen Keyser, Gerhard van der Horst, Liana Maree

**Affiliations:** Comparative Spermatology Laboratory, Department of Medical Bioscience, Faculty of Natural Sciences, University of the Western Cape, Private Bag X17, Bellville 7535, South Africa; skeyser@uwc.ac.za (S.K.); gvdhorst7@gmail.com (G.v.d.H.)

**Keywords:** progesterone, myo-inositol, dopamine, prolactin, sperm subpopulations, functional characteristics, double density gradient centrifugation, CASA, high motile, low motile

## Abstract

Considering the challenges surrounding causative factors in male infertility, rather than relying on standard semen analysis, the assessment of sperm subpopulations and functional characteristics essential for fertilization is paramount. Furthermore, the diagnostic value of sperm interactions with biological components in the female reproductive tract may improve our understanding of subfertility and provide applications in assisted reproductive techniques. We investigated the response of two sperm motility subpopulations (mimicking the functionality of potentially fertile and sub-fertile semen samples) to biological substances present in the female reproductive tract. Donor semen was separated via double density gradient centrifugation, isolated into high (HM) and low motile (LM) sperm subpopulations and incubated in human tubal fluid (HTF), capacitating HTF, HD-C medium, progesterone, myo-inositol, dopamine and prolactin. Treated subpopulations were evaluated for vitality, motility percentages and kinematic parameters, hyperactivation, positive reactive oxygen species (ROS), intact mitochondrial membrane potential (MMP) and acrosome reaction (AR). While all media had a significantly positive effect on the LM subpopulation, dopamine appeared to significantly improve both subpopulations’ functional characteristics. HD-C, progesterone and myo-inositol resulted in increased motility, kinematic and hyperactivation parameters, whereas prolactin and myo-inositol improved the LM subpopulations’ MMP intactness and reduced ROS. Furthermore, progesterone, myo-inositol and dopamine improved the HM subpopulations’ motility parameters and AR. Our results suggest that treatment of sub-fertile semen samples with biological substances present in follicular fluid might assist the development of new strategies for IVF treatment.

## 1. Introduction

The aim of male fertility evaluations is to detect possible abnormalities in reproductive system and to determine which cases may require assisted reproductive technologies (ART) as treatment for low sperm count or quality [1,2]. Despite the increased use of ART, 68.5% of in vitro fertilization (IVF) cycles do not result in a live birth [3] and overall pregnancies per aspiration have only increased by 0.2% as compared to rates in 2012 [4]. Consequently, alternative approaches and interventions that may improve not only conventional sperm parameters, but also the genetic material of spermatozoa and thereby ART success rates [2], should be explored. Ideally, such approaches and interventions would be applicable in cases where fertility issues can be corrected without the need of costly ART cycles.

Spermatozoa must be functionally and structurally intact to traverse the female reproductive tract, respond to various chemo- and thermotactic signals and ultimately achieve successful fertilization. As a result, basic semen parameters alone cannot predict the fertility potential of ejaculates; additional functional tests should be explored, including how spermatozoa respond to various biological substances found within the female reproductive tract [5]. The functionality of spermatozoa is closely associated with posttranslational modifications initiated by secondary messengers, as these cells are transcriptionally and translationally silent [6,7,8,9]. Mature spermatozoa with a high fertility potential typically exhibit progressive motility, normal morphology, viability, ability to undergo capacitation (hyperactivation and acrosome reaction), intact mitochondrial membranes (MMP), normal levels of reactive oxygen species (ROS) and intact DNA [10]. 

Due to the fact that signaling events in human spermatozoa remain poorly understood, along with the identities of many molecules that stimulate them [11], the current study aimed to establish the unique and collective biological effects of various components present in follicular fluid (FF) on human sperm motility subpopulations. It is well known that FF contains some metabolites critical for oocyte growth and development, forming a microenvironment that plays a key role in fertilization, implantation and early embryo development [12]. In the female reproductive tract and FF, substances such as progesterone, myo-inositol, dopamine, and prolactin are assisted by cyclic AMP (cAMP)/cAMP dependent protein kinase A (PKA) pathways to emanate the physiological changes that allow spermatozoa to fertilize oocytes [7,13,14,15,16,17,18,19,20,21]. 

In human spermatozoa, progesterone-induced Ca^2+^ influx is mediated through CatSper channels and its associated proteins, thereby regulating capacitation, hyperactivation and the acrosome reaction [14,19,22,23]. Moreover, it has been proposed that only capacitated human spermatozoa are able to participate in chemotactic swimming by using progesterone gradients in close proximity to the oocyte [18]. In human epididymal and testicular spermatozoa, CatSper currents showed sensitivity to progesterone early in sperm development and peaks when spermatozoa are ejaculated [24]. 

Myo-inositol is involved in numerous biological processes such as protein binding to the cell surface, cell signaling, vesicle trafficking, membrane excitability, regulation of channel opening and intracellular Ca^2+^ signaling [25]. Myo-inositol is an important precursor for signaling pathways which form secondary messengers that modulate protein phosphorylation and intracellular Ca^2+^ concentrations [7]. As such, myo-inositol present in the female reproductive tract plays a vital role in improving sperm quality, including motility, viability and cholesterol efflux, which regulates sperm capacitation [26,27]. 

High concentrations of catecholamines have been detected in both human semen and the oviduct [13,28]. Dopamine type 1-like receptors (DRD1) activate adenylyl cyclase which results in increasing cAMP accumulation and activating PKA, whereas dopamine type 2-like receptors (DRD2) inhibits adenylyl cyclase, thereby decreasing PKA activity [13]. Dopamine appears to act physiologically as a regulator of viability, fertility and sperm motility [13,29]. Dopamine-induced activation of DRD2 in boar sperm seems to have a beneficial effect on cell viability [16]. 

Prolactin plays an important role in female reproduction but prolactin receptors (PRL-R) have been detected in male reproductive organs, with a positive modulatory effect on various aspects of testicular function [30]. Pro-survival mechanisms of prolactin on spermatozoa have been suggested; however, it remains unclear whether prolactin has a stimulatory, inhibitory or no measurable effect on sperm capacitation [6].

Due to practical and ethical considerations, information about human sperm interaction with oviductal and follicular fluid contents is scarce in comparison with rodents and other mammalian species [18]. In this study, a comprehensive set of parameters and advanced statistical approaches were utilized to investigate and compare functional and structural sperm characteristics between two sperm motility subpopulations (HM: high motile and LM: low motile) after exposure to various concentrations of progesterone, myo-inositol, dopamine and prolactin. We hypothesize that these biological substances found within the female reproductive tract will differentially improve/change sperm function of the two sperm subpopulations. These subpopulations were used as a direct mimicking model for “fertile” and “sub-fertile” semen samples [5,31,32], respectively, to determine whether the selected substances could enhance sperm quality and further be utilized in ART to improve sub-fertile samples and possibly result in successful fertilization.

## 2. Materials and Methods

### 2.1. Sample Collection and Standard Semen Analysis

One hundred and sixty human semen samples from forty-four donors were obtained via masturbation as part of a donor program (Division of Medical Physiology, Department of Biomedical Sciences, Stellenbosch University and Department of Medical Bioscience, University of the Western Cape) after two to three days of sexual abstinence. Donor samples were incubated permitting liquefaction (30–60 min at 37 °C in a 5% CO_2_ regulated incubator) and subsequently processed as recommended by the World Health Organization (WHO) [33]. Semen volume, pH and viscosity were assessed, as well as several sperm parameters, including total motility, progressive motility, sperm concentration, total number of spermatozoa, mucus penetration, vitality and morphology (analyzed with Sperm Class Analyser^®^ (SCA^®^) computer-aided sperm analysis (CASA) system, version 6.2 (Microptic S.L., Barcelona, Spain)). A minimal cut-off point for percentage total sperm motility of 15% was used, as the basis of the study was to determine how various concentrations of media can improve different qualities of spermatozoa. The study was approved by the ethical boards of the University of the Western Cape (code BM20/9/14), and Stellenbosch University (code N14/06/074). The Helsinki Declaration [34] governing research on humans has been adhered to and each human donor gave written consent.

### 2.2. Preparation of Sperm Subpopulations

Two motility sperm subpopulations (highly motile spermatozoa, HM; less motile spermatozoa, LM) were prepared from semen samples through double density gradient centrifugation (DGC) with the use of AllGrad^®^ 90/45% and AllGrad Wash^®^ (Delfran, Johannesburg, South Africa). Equal volumes (300 µL) of semen and preheated (37 °C) density gradient 90–45% were layered in an Eppendorf tube, and centrifuged at room temperature (RT) for 20 min at 500× *g*. Resultant top coats consisting of seminal plasma were discarded, and remaining intermediate layers and bottom pellets were separated into Eppendorf tubes as top (LM subpopulation) and bottom (HM subpopulation) subpopulations, respectively. Subpopulations were re-suspended in 300 µL AllGrad Wash^®^ and centrifuged at 500× *g* for 10 min. Washed pellets were resuspended in the various media and final sperm concentrations adjusted to 15–25 × 10^6^/mL before being assessed.

### 2.3. Preparation of Media

Four of the five different media (containing progesterone, myo-inositol, dopamine and prolactin) were prepared at various concentrations in capacitating HTF (constituted of non-capacitating HTF supplemented with 0.105 g NaHCO_3_, 1.1915 g HEPES and 0.6 mL NaOH), whereas non-capacitating HTF (HTF) and capacitating HTF (CAP) [35] functioned as the negative and positive controls, respectively. All media were void of 1% human serum albumin (HSA) supplementation, as samples were not exposed to extensive incubation hours and the study aimed to determine the individual effects of the media alone on sperm function. Due to the negative effect that time has on sperm functionality, selected media was grouped together and groups were assessed on separate samples and occasions; however, controls were included throughout. To this effect, HD-C, myo-inositol and progesterone were grouped and analyzed together, after which dopamine concentrations and prolactin concentrations were respectively grouped and analyzed individually for each group. Furthermore, due to time constraints, the investigation of functional parameters was additionally grouped and analyzed on different samples and occasions as follows: vitality was assessed on 20 samples; motility and hyperactivation were assessed together on 20 samples; and MMP, ROS and AR were assessed together on 20 samples. For the dopamine group, vitality, motility and hyperactivation could be analyzed together on the same 20 samples due to fewer concentrations being investigated.

#### 2.3.1. HD-C Medium

HD Sperm Capacitation™ medium (HD-C, HDSC-0005) purchased from Delfran Pty Ltd. (Delfran, Johannesburg, South Africa) comprised of a mixture of NaCl, KCl, MgSO_4_.7H_2_O, KH_2_PO_4_, Na_2_HPO_4_, NaHCO_3_, CaCl_2_, glucose, progesterone and myo-inositol for inducing hyperactivation in spermatozoa.

#### 2.3.2. Progesterone

Progesterone in dimethyl sulfoxide (DMSO; Sigma-Aldrich, Cape Town, South Africa) was prepared in CAP to yield three working solutions of 1.98 µM, 3.96 µM and 19.8 µM (volume/volume). Selected concentrations were based on previous investigations in addition to physiological concentrations in the female reproductive tract [36,37,38].

#### 2.3.3. Myo-Inositol

Myo-inositol (MW = 180.16 g/mol) was weighed out and dissolved in CAP (mass/volume) to yield a working solution of 11 mM. This selected concentration was based on previous investigations in addition to physiological concentrations in the female reproductive tract [36,37,38]. 

#### 2.3.4. Dopamine

Dopamine hydrochloride (H8502, Sigma-Aldrich, Cape Town, South Africa) was weighed out and dissolved in CAP (mass/volume) to yield a stock solution of 1 mM dopamine, which was further diluted to three working solutions of 20 nM, 100 nM and 1 µM (volume/volume). Selected concentrations were adjusted from previous investigation and physiological concentrations found within the female reproductive tract [13,29,39,40].

#### 2.3.5. Prolactin

Hydrolyzed Prolactin (L7009, Sigma Aldrich, Cape Town, South Africa) was prepared in 4 mM HCl (mass/volume) to a stock solution of 10 µg/mL and was diluted in CAP to yield four working solutions of 50 ng/mL, 100 ng/mL, 250 ng/mL and 500 ng/mL. Selected concentrations were adjusted from previous investigations [6,41,42,43].

### 2.4. Viscosity

Using the viscosity evaluation technique described by Rijnders et al. [44], 3 µL semen aliquots were loaded into preheated (37 °C) four-chamber, 20 μm-depth Leja slides (Leja Products B.V., Nieuw Vennep, The Netherlands) and the filling time was recorded in seconds (s). Viscosity in centipoise (cP) was subsequently determined by using the following equation:y = 0.34x + 1.34 (1)
where y = viscosity in cP and x = filling time in s.

### 2.5. Sperm Morphology

Semen aliquots (300 µL) were centrifuged in AllGrad Wash^®^ at RT for 20 min at 500× *g* and subsequent pellets re-suspended in HTF. Morphology smears were prepared (15 µL) and dried slides stained with SpermBlue^®^ fixative and stain mixture (Microptic S.L., Barcelona, Spain) as described by van der Horst and Maree and modified by Microptic [45,46]. Coverslips were mounted with DPX mounting medium (Sigma Aldrich, Cape Town, South Africa) and 100 spermatozoa were analyzed with the Morphology module of the SCA^®^ using brightfield optics, an acA1300-200uc camera, a blue filter and a 60× objective on a Nikon Eclipse 50i microscope (IMP, Cape Town, South Africa).

### 2.6. Sperm Vitality

All reagents and equipment were heated to 37 °C prior to use. Subpopulation vitality smears were prepared after 5 and 30 min incubation in the various media, with the additional time point of 60 min for prolactin. Semen samples and treated sperm subpopulations were stained in suspension with BrightVit (Microptic S.L., Barcelona, Spain), and smears prepared following the staining and preparation technique proposed by Microptic [46]. Dried vitality smears were mounted with a coverslip using DPX mounting medium and viewed with brightfield optics and a 100× oil immersion objective. Percentages of viable spermatozoa was calculated after at least 100 spermatozoa per slide were manually assessed.

### 2.7. Sperm Motility, Concentration and Mucous Penetration

Total sperm motility, progressive motility, concentration and mucus penetration were assessed with the Motility module of the SCA^®^ and data captured with a Basler acA1300-200uc digital camera (Microptic S.L., Barcelona, Spain) attached to a Nikon Eclipse 50i microscope with a 10x positive phase contrast objective, green filter and heated stage. Preheated (37 °C) four or eight-chamber, 20 μm-depth Leja slides were loaded with 2–3 µL of semen or treated subpopulations and at least two fields with 200 motile spermatozoa were analysed at 50 frames per second (f/s). 

After incubation (5, 30 and 60 min) in the different media, motility percentages assessed for treated subpopulations included total motility, progressive motility, rapid-, medium- and non-progressive motility, as well as rapid-, medium- and slow-swimming spermatozoa. In addition, eight kinematic parameters, including curvilinear velocity (VCL), straight-line velocity (VSL), average path velocity (VAP), linearity (LIN), straightness (STR), wobble (WOB), amplitude of lateral head displacement (ALH) and beat cross frequency (BCF), were recorded for the average motile sperm population as well as various subpopulations for progressiveness. ALH was measured as half the width of the VCL track and not as the full VCL wave or doubling of riser values (risers’ method) as described by Mortimer [35,47]. Kinematic parameter cut-off values for mucus penetration were VAP > 25 µm/s, STR > 80% and 7.5 µm < ALH > 2.5 µm [48,49].

### 2.8. Hyperactivation

Applying the flush technique described by van der Horst et al. [50], each chamber of a 20 μm-depth, eight-chamber Leja slide was loaded with 0.5 µL sperm preparation (HM or LM sperm subpopulations suspended in HTF) and flushed with 1.5 µL of preheated media as mentioned in Section 2.3. Percentages of hyperactivation (using cut-off values: VCL > 150 μm/s; LIN < 50%; ALH > 7 μm (3.5 for SCA^®^)) [47] of at least 200 motile spermatozoa was assessed after 5, 15, 30, 45 and 60 min exposure to each treatment, for each sperm subpopulation using the Motility module of SCA^®^ and equipment describe in the previous section. 

### 2.9. Reactive Oxygen Species

Subpopulations were incubated for 30 min in media mentioned in Section 2.3, after which 20 µM Dihydroethidium (DHE, excitation = 518 nm and emission = 605 nm, Molecular Probes, Eugene, OR, USA) was used to detect spermatozoa positive for reactive oxygen species (ROS). Treated subpopulations were stained in the dark for 15 min in suspension (180 µL) with 20 µL of DHE at 37 °C. Following incubation, 5–10 µL of suspension was placed on a clean slide with a coverslip, and immediately analyzed using a 100× oil immersion objective and triband filter (MXU440) (excitation wavelengths: 457 nm = blue, 530 nm = green and 628 nm = red) on a Nikon Eclipse 50i fluorescence microscope. Percentages of spermatozoa positive for ROS were calculated after manual assessment of at least 100 spermatozoa.

### 2.10. Mitochondrial Membrane Potential (ΔΨm)

The protocol of the Mitochondria Staining Kit (CS0390, Sigma Aldrich, Cape Town, South Africa) to assess mitochondrial membrane potential (MMP) was optimized for this specific study. Subpopulations were incubated for 30 min in the media mentioned in Section 2.3, followed by staining (1:1 ratio) in the dark at 37 °C for 20 min in suspension (200 µL) with the MMP staining solution (160 µL dH_2_O, 40 µL JC-5 buffer and 1 µL frozen MMP 200x stock solution). After incubation, suspensions were centrifuged at 500× *g* for 5 min at 5–7 °C and washed pellets re-suspended in 200 µL JC-1 buffer (80 µL JC-5 buffer and 320 µL dH_2_O), prepared and cooled on ice before use. Suspensions were centrifuged for a second time as described above and subsequent pellets were re-suspended in the remaining 200 µL JC-1 buffer. A single drop of 5–10 µL of suspension was placed on a clean slide and covered with a coverslip. Slide preparations were immediately analyzed using the same equipment as described in Section 2.9, and the percentage of spermatozoa with intact MMP calculated after manual assessment of at least 100 spermatozoa. 

### 2.11. Acrosome Reaction

The acrosome reaction was determined with the use of the FluAcro protocol described by Microptic [46]. Subpopulations were incubated at 37 °C for 3 h in preheated (37 °C) capacitating medium (1 mL HAMS-F10 and 0.03 g HSA; Sigma-Aldrich, Cape Town, South Africa). After incubation, subpopulations were treated with 10 µL of each of the following media: 1 mM Ca-ionophore made up in DMSO (yielding a final concentration of 0.01 mM); capacitating and non-capacitating HTF; HD-C; progesterone; myo-inositol; dopamine; and prolactin. Samples were incubated in the media for 15 min after which reactions were terminated with 100 µL 70% ethanol. Two 5 µL drops of each suspension were placed on a clean slide and left to air dry before fixation in 95% ethanol (United Scientific, Cape Town, South Africa) at 4 °C for 30 min. Fixed slides were stained in a dark room for 30–40 min with 80 µL fluorescein isothiocyanate-labelled peanut agglutinin (FITC-PNA; Sigma-Aldrich, Cape Town, South Africa) on each drop, and dipped twice in dH_2_O to remove excess stain. Slides were subsequently counterstained for 7 min with 5 µL Hoechst (H33258, Sigma-Aldrich, Cape Town, South Africa) on each drop, followed by destaining in dH_2_O, and further left to air dry before viewing. The acrosome status of at least 100 spermatozoa per sample was manually assessed using a 40× quartz fluorescence objective and a triband fluorescence filter (MXU440) on a Nikon Eclipse 50i epi-fluorescence microscope.

### 2.12. Statistical Analysis

MedCalc statistical software version 14.8.1 (Mariakerke, Gent, Belgium) was used to calculate basic summary statistics, and results are presented as mean ± standard deviation in all the tables. Treated sperm subpopulations were compared using the Student’s t-test or the Mann–Whitney test when normal distribution was lacking. Where applicable, the one-way analysis of variance (ANOVA) for parametric distributions or the Kruskal–Wallis test for non-parametric distributions were used to compare various subpopulations, time points and media. Significance was determined at a level of *p* < 0.05 with the Student–Newman–Keuls post hoc test. Additional analyses, such as multifactorial ANOVA and the creation of multivariable charts, were performed with Statgraphics^®^ Centurion XVII (Statgraphics Technologies, Inc., The Plains, VA, USA). More advanced statistics were calculated by performing a mixed model repeated measures ANOVA with STATISTICA, version 10 (StatSoft Inc., Tulsa, OK, USA). Subpopulations, media and time were used as fixed factors and samples as the random factors. The Fisher LSD was used for the post hoc test and reports based on third-order effects (interactions amongst subpopulations, media and time). If third-order effects were absent, reports were based on interactions between fixed factors (subpopulations and media, subpopulations and time or media and time). If none of the fixed factors (subpopulations, media or time) were included in interactions, only the main effects were reported.

## 3. Results

### 3.1. Bubble Diagrams and Quantitative Results

To indicate the comprehensive effects of the media on sperm subpopulations, and collectively, with media as a fixed factor, the results are summarized and displayed as bubble diagrams. Figure 1 displays the overall effects of each medium on functional parameters, including vitality, mitochondrial membrane potential (MMP), positive reactive oxygen species (ROS), hyperactivation (HA) and acrosome reaction (AR). Illustrated in Figure 2 and Figure 3, respectively, are the effects of each medium on motility percentages and average kinematic parameters. Individual functional parameters are considered in more detail under the subsequent subheadings; however, bubble diagrams will be referred back to. Results for sperm functional parameters are depicted as figures in the text but detailed tables with the actual data are presented as Appendix A and will be referred to as such. 

### 3.2. Standard Semen Analysis

Average semen parameters of donor samples used in this investigation are displayed in Table 1. Mean values ranged above lower reference values as recommended by the WHO laboratory manual [33], with the exclusion of progressive motility falling in the 2.5th percentile.

### 3.3. Vitality

High motile (HM) subpopulations had significantly (*p* < 0.05, *p* < 0.01 and *p* < 0.001) higher vitality percentages as compared to low motile (LM) subpopulations (Figure 4A–C). However, after 5 min incubation, 20 nM dopamine improved the LM vitality percentages (Figure 1) to the extent that significant (*p* = 0.1653) differences between subpopulations were eliminated (Figure 4B). No significant differences among media was observed for individual subpopulations’ vitality percentages. Furthermore, only HTF vitality percentages of the HM subpopulations were significantly (*p* = 0.01) lower at 30 min as compared to 5 min incubation (Figure 4B).

To determine whether combined factors (subpopulations, time and media) had an interaction on vitality percentages, data were analyzed in a mixed model repeated measures ANOVA. A significant effect (*p* < 0.01) was only observed for media as fixed factor, as illustrated in Figure 5A,B. For individual effects of media, HD-C (*p* < 0.01), progesterone (1.98 µM, *p* = 0.03; 3.96 µM, *p* = 0.03 and 19.8 µM, *p* = 0.02; Figure 5A) and prolactin (50 ng/mL, 100 ng/mL and 250 ng/mL, *p* < 0.01; Figure 5B) significantly improved percentages of viable spermatozoa as compared to HTF. In addition, myo-inositol (*p* < 0.01) significantly improved vitality as compared to both HTF and CAP. Prolactin appeared to display a possible dose-dependent trend as 100 ng/mL resulted in significantly (*p* < 0.01) higher vitality percentages as compared to 500 ng/mL (Figure 5B).

### 3.4. Motility and Kinematic Parameters

Compared to the LM subpopulations, the HM subpopulations frequently displayed significantly higher percentages for various motility parameters at 5, 30 and 60 min incubation in HTF (Appendix A). After incubation in various media, motility percentages were improved in the LM subpopulations to the extent that significant differences between subpopulations were eliminated (Figure 2). This was mainly observed for motility speed groups (rapid, medium and slow) and progressivity speed groups (RP and MP) for various media, but predominantly for higher concentrations of dopamine (100 nM and 1 µM) and prolactin (100 ng/mL and 500 ng/mL).

For the HM subpopulations, various media improved motility percentages as compared to HTF (Appendix A). Percentages of rapid and RP speeds were increased by CAP, progesterone, myo-inositol and dopamine—whereas only CAP and prolactin increased percentages of total motility (Figure 2). For the LM subpopulations, prolactin increased both total motility and slow speed groups, while all media improved progressive, RP, MP, rapid and medium speed groups of this subpopulation (Figure 2). Whereas higher concentrations of dopamine (1 µM) and progesterone (3.96 µM) displayed larger stimulatory effects on motility percentages of subpopulations, prolactin varied in its effects among concentrations by displaying a possible biphasic or dose-dependent response (Appendix A).

No interactions of media, subpopulations and time points on motility percentages were revealed in a mixed model repeated measures ANOVA; however, media as fixed factor had a significant (*p* < 0.01) effect on various motility percentages (see Figure 6A–C, Figure 7A–C and Figure 8A–C for examples). Compared to HTF, all media increased the percentages of progressive motility (Figure 2, Figure 6A, Figure 7A and Figure 8A), rapid and RP speed groups (Figure 2, Figure 6B, Figure 7B and Figure 8B) and MP speed groups (Figure 2, Figure 6C, Figure 7C and Figure 8C), while decreasing the percentages of slow speed groups (Figure 2). In contrast, CAP and prolactin appeared to increase the percentage of medium speed groups (Figure 2). A dose-dependent response for prolactin was observed again as the highest concentration (500 ng/mL), often presented with decreased motility percentages as compared to lower concentrations (100 ng/mL and 250 ng/mL).

### 3.5. Hyperactivation

The HM subpopulation displayed significantly higher values for percentage hyperactivation as compared to the LM subpopulations at most time points when incubated in HTF (Appendix A). Exposure to different concentrations of the selected biological substances generally induced a higher percentage hyperactivation in both subppulations over time (Figure 1). However, the media had a more pronounced effect on the LM subpopulations which resulted in the elimination of the differences between the two subpopulations (Figure 1) at most time points (Appendix A). Increased concentrations of the substances did not necessarily have the same effect on the two subpopulations. For instance, dopamine significantly increased hyperactivation in the HM subpopulations as compared to HTF; however, higher dopamine concentrations appeared to increase hyperactivation in the LM subpopulations, as compared to HTF and CAP (Appendix A). Prolactin also resulted in an increase in percentage hyperactivation of both subpopulations as compared to HTF; however, less variation between concentrations and time points were observed in the LM subpopulations as was found in the HM subpopulation (Appendix A). 

Individual fixed-factor effects as well as interactions between fixed factors as were highlighted with a mixed model repeated measures ANOVA are displayed in Figure 9, Figure 10 and Figure 11. Both media and time as fixed factors had significant effects on hyperactivation percentages after exposure to HD-C, progesterone, myo-inositol (Figure 9) and prolactin (Figure 11). While CAP, progesterone, myo-inositol and prolactin induced hyperactivation, exposure to HD-C and 3.96 µM progesterone (Figure 9A) and all concentrations of prolactin (Figure 11A) resulted in significantly higher percentages. An increase in percentage hyperactivation was observed from 5 min to 15 min after exposure to media, where after a significant decline was observed between 30–45 min, indicating an early stimulatory effect of media on hyperactivation (Figure 9B and Figure 11C).

Fixed factor interactions were found for “subpopulations and media” after exposure to dopamine and prolactin as well as for “subpopulations and time” after exposure to dopamine (Figure 10A,B and Figure 11B). While exposure to all concentrations of dopamine resulted in significantly higher percentages hyperactivation (compared to HTF and CAP), and 1 µM dopamine resulted in the highest hyperactivation percentages in both subpopulations, there was no difference in the percentage hyperactivation induced by 100 nM and 1 µM dopamine in the LM subpopulations (Figure 10A). A similar trend was seen after prolactin exposure, with significantly higher percentages of hyperactivation found compared to HTF in the HM subpopulations, as well as compared to HTF and CAP in the LM subpopulations; however, the highest concentrations of prolactin (250 ng/mL and 500 ng/mL) did not induce significantly higher percentages hyperactivation (Figure 11B). In terms of exposure time (Figure 10B), the LM subpopulations displayed a delayed or prolonged response to dopamine (insignificant increase in hyperactivation from 5 to 30 min with significant decrease at 60 min), whereas the HM subpopulations displayed a rapid response (increase in hyperactivation from 5 to 15 min with significant decrease at 45 min). Illustrated in Figure 10C, a fixed-factor interaction was also found for “media and time” after exposure to dopamine. HTF, CAP and dopamine resulted in an increased percentage hyperactivation from 5 to 30 min.

### 3.6. Reactive Oxygen Species

Significantly higher percentages of ROS positive spermatozoa (*p* < 0.05 and *p* < 0.01) were recorded for the LM subpopulations compared to the HM subpopulations after exposure to all media (Figure 12A–C). However, exposure to 100 nM and 1 µM dopamine, as well as 50 ng/mL, 100 ng/mL and 250 ng/mL prolactin reduced ROS levels of the LM subpopulations which eliminated these differences between subpopulations (Figure 1). In addition, only prolactin in the LM subpopulations significantly decreased ROS levels (*p* = 0.008) as compared to HTF (Figure 12C). After data for subpopulations was pooled in a multifactorial ANOVA, significant interactions between media and ROS percentages were seen for HD-C, progesterone and myo-inositol (*p* = 0.0059), as well as for prolactin (*p* = 0.002), but not dopamine (Appendix A). 

### 3.7. Mitochondrial Membrane Potential

HM subpopulations had significantly higher percentages of spermatozoa with intact MMP (*p* < 0.05, *p* < 0.01 and *p* < 0.001) compared to the LM subpopulations after exposure to all media (Figure 13A–C). However, 1 µM dopamine, as well as 50 ng/mL, 250 ng/mL and 500 ng/mL prolactin were able to maintain and increase MMP intactness in the LM subpopulations to the extent that differences between subpopulations were eliminated (Figure 1). No significant differences between media were seen in the HM subpopulations; however, in the LM subpopulations, myo-inositol (*p* = 0.048) as well as 50 ng/mL, 250 ng/mL and 500 ng/mL prolactin (*p* = 0.012) all significantly maintained higher percentages of intact MMP as compared to HTF (Figure 13A–C). Once data were pooled and analyzed in the multifactorial ANOVA, a significant interaction was observed between media and percentages of MMP intactness (Appendix A). Compared to HTF, 500 ng/mL prolactin maintained significantly higher MMP percentages (*p* = 0.0037). Myo-inositol also significantly maintained a higher percentage MMP intactness (*p* = 0.0153) as compared to both HTF and CAP. 

### 3.8. Acrosome Reaction

The LM subpopulations revealed significantly higher percentages of spontaneous and induced acrosome reaction compared to the HM subpopulations after treatment with HTF, CAP, HD-C, progesterone (Figure 14A) and prolactin (Figure 1 and Figure 14C). In contrast, the HM subpopulations had significantly higher percentages of Ca-ionophore induced acrosome reaction (Figure 14A–C) compared to the LM subpopulations. Nonetheless, progesterone, myo-inositol and dopamine increased the percentage of acrosome reacted spermatozoa in the HM subpopulations (Figure 1 and Figure 14A,B). In the LM subpopulations, only myo-inositol significantly increased the percentage of acrosome reacted spermatozoa as compared to HTF (Figure 14A). 

For both subpopulations, Ca-ionophore consistently obtained significantly higher percentages of acrosome reacted spermatozoa (*p* < 0.001) as compared to all other media used (Figure 14A–C, Appendix A). Once data were pooled in a multifactorial ANOVA, significant interactions between media and acrosome reaction were observed. Compared to HTF and CAP, progesterone and myo-inositol significantly increased the percentage acrosome reacted spermatozoa (*p* < 0.001) (Appendix A). Compared to HTF, 100 nM and 1 µM dopamine significantly increased the percentage acrosome reacted spermatozoa (*p* < 0.001) (Appendix A).

## 4. Discussion

This study investigated the effects of specific biological substances found in follicular fluid (FF) on two human sperm motility subpopulations. Our results confirm that HM and LM subpopulations are diverse in their structural and functional characteristics and indicate that these subpopulations respond differently after short-term exposure to various concentrations of the biological substances tested. While the HM subpopulations’ functional status was often not significantly affected by such exposure, progesterone, myo-inositol and dopamine improved the HM subpopulations’ motility parameters and acrosome reaction. In contrast, all the biological substances investigated were capable of improving the LM subpopulations’ reduced functional capabilities to some extent (as compared to HM subpopulations) and, in many instances, such improvements eliminated the initial differences found between the two subpopulations in HTF. 

### 4.1. Progesterone

Progesterone in FF varies in concentrations throughout the female reproductive tract, and is well known to effect spermatozoa [21] due to progesterone-sensitive CatSper Ca^2+^ channels located near the principal piece of the flagellum [21,51,52,53,54]. Ghanbari et al. [53] observed a reduction in sperm vitality after CatSper channels of progesterone-stimulated sperm were inhibited. In the current study, progesterone had no effect on percentage vitality of individual sperm subpopulations, which agrees with the findings reported by Contreras et al. [55]. However, with media considered as a fixed factor, exposure to progesterone increased the overall sperm vitality percentages compared to the control medium. 

Progesterone increased the rapid speed and rapid progressivity (RP) groups, with increased percentages reported for both subpopulations. In accordance with our results, Ghanbari et al. [53] found that progesterone increased progressive motility, whereas Pujianto et al. [56] reported that progesterone displayed a dose-dependent response with increased motility percentages at higher concentrations (500 ng/mL). In the current study, 3.96 µM progesterone seems to have the largest stimulatory effect on both subpopulations’ motility percentages as well as collectively with media as fixed factor. Progesterone increased the velocity kinematics of the RP and MP groups, but linear kinematics decreased. Calogero et al. [57] similarly reported that progesterone increased BCF and VCL, but decreased LIN and STR of motile spermatozoa. Furthermore, the observed improvements in the LM subpopulations’ motility parameters may be a result of either intracellular pH or proper balancing of intracellular ions, and not necessarily activation of progesterone CatSper Ca^2+^ influx [58]. 

Progesterone is well documented to induce hyperactivation and elicits different effects at varying concentrations [53,54,57,59,60,61]. Similarly, we observed that exposure to 3.96 µM progesterone induced higher hyperactivation percentages compared to 1.98 µM and 19.8 µM progesterone. Whereas typical steroid signaling pathways are time-consuming due to hormones binding to genomic receptors, progesterone bypasses this process by binding to sperm plasma membrane receptors and causes a rapid Ca^2+^ influx that is sustained for several minutes [52,57]. This effect of progesterone is in agreement with our findings, with an increased percentage hyperactivation observed after 5 min of exposure. However, time differences for subpopulations were evident, with the LM subpopulations exhibiting a small, gradual increase and decrease in percentage hyperactivation after progesterone exposure, while the HM subpopulations revealed a sudden increase and decrease in this sperm parameter. Progesterone-induced Ca^2+^-influx followed by Ca^2+^ oscillations in the intracellular compartment could be responsible for the observed motility and hyperactivation patterns in the HM subpopulations, by prolonging motility regulation and assisting in repeated alteration of activated and hyperactive motility [61]. Genomic errors at CatSper subunit loci are reported in infertile men; it has been suggested this prevents functional channel expression and completely compromises fertilization in vivo and in vitro [62]. This could be a plausible causative effect of the lower percentage of hyperactivation and only small increases found in the LM subpopulations as compared to the HM subpopulations. 

The physiological activation of AR by progesterone is well documented for humans, but less so between different sperm subpopulations [53,57,60,63]. Our results indicate that progesterone only induced AR in the HM subpopulations, but values remained in the range of 20–50%, and thus significantly lower than the 60–75% AR reported after Ca-ionophore exposure. Similarly, Tantibhedhyangkul et al. [60] reported that despite progesterone-induced Ca^2+^ increases, less than 50% of spermatozoa actually undergo AR. The seemingly inability of progesterone to rapidly induce higher hyperactivation and AR percentages in the LM subpopulations could be related to slowness or deficiency in the formation of Ca^2+^ signaling domains. To successfully achieve capacitation by CatSper, a quadrilateral arrangement of four pore-forming α-subunits needs to develop, which structurally organizes unique Ca^2+^ signaling domains along the flagellum [58,64]. The absence or alteration of genes that prevent the formation of this heterotetrametric CatSper complex will thereby prevent capacitation [58,64].

Even though Tantibhedhyangkul et al. [60] reported a progestin-dependent increase in MMP after 90 min incubation, the MMP levels of both subpopulations did not significantly change after 30 min progesterone exposure in the current study—the difference possibly being related to exposure time. It was speculated that increased progesterone concentrations in the female reproductive tract may stimulate mitochondrial progesterone receptors (PR-M), increasing mitochondrial ATP [60]. Furthermore, Fan et al. [65] reported that progesterone reduced ROS production as well as balanced MMP in spermatozoa. Our results indicated no effect of progesterone exposure on ROS levels in the two subpopulations; however, when media was considered as a fixed factor, progesterone significantly reduced ROS positive spermatozoa. 

### 4.2. Myo-Inositol

As a precursor to secondary messengers in cellular signal transduction systems, myo-inositol plays an essential role in morphogenesis, cytogenesis, membrane formation/growth and lipid production [66,67,68]. Dog sperm with myo-inositol supplementation reportedly had higher vitality percentages as compared to controls, which is consistent with our results [68]. Myo-inositol displayed the greatest effect on preserving sperm vitality in both the LM subpopulation and collectively with media as fixed factor. The ability to maintain and preserve sperm vitality may be a result of myo-inositol’s role in lipid production, thereby maintaining intact plasma membranes [68]. 

Similar to progesterone, myo-inositol generally improved sperm motility, but had a more substantial effect on the LM subpopulations’ motility parameters (rapid and RP speed percentages) as well as kinematic parameters (MP and NP speed groups). According to Governini et al. [69], treatment with myo-inositol significantly increased sperm motility and oxygen consumption in oligoasthenozoospermic patients. In addition, Artini et al. [70] observed an increase in progressive motility of both normospermic and oligoasthenospermic semen after treatment with myo-inositol (15 μL/mL). These observations in LM subpopulations and infertile/subfertile males after myo-inositol exposure may be indicative of its ability to increase oxidative phosphorylation efficiency and ATP production by the mitochondria [71]. 

Myo-inositol induced hyperactivation and AR in both subpopulations, as well as when media was considered as a fixed factor. By activating phospholipase-C, myo-inositol results in production of inositol triphosphate, which opens Ca^2+^ channels and regulates intracellular Ca^2+^ stores in the sperm plasma membrane, mitochondria, acrosome and neck. Ultimately, myo-inositol thus indirectly stimulates oxidative mechanisms and ATP production, which improve mitochondrial function, prevent apoptosis and assists sperm capacitation [69,72,73]. 

Furthermore, myo-inositol also improved MMP of the LM subpopulation and had the largest effect on improving MMP and decreasing ROS percentages when media was considered as a fixed factor. It is reported that elevated ROS levels compromise sperm quality due to lipid peroxidation and loss of membrane fluidity, mitochondrial dysfunction, altered morphology, reduced vitality and other changes that result in the failure to fertilize [73]. Supplementation with myo-inositol thus seems to be beneficial, since it preserves vitality and MMP function, reduces and controls ROS levels, improves ATP production and reduces oxidative stress that alters capacitation. As oxidative stress has been observed in 30–80% of infertile patients, multi-antioxidant supplementation is an effective treatment that helps to improve male fertility parameters [73]. 

### 4.3. HD-C Medium

HD-C medium is a sperm capacitation medium containing both progesterone and myo-inositol. Exposure of sperm subpopulations to HD-C resulted in similar effects on preserving sperm vitality as progesterone and myo-inositol when media was considered as a fixed factor. HD-C exposure also displayed a similar pattern to progesterone and myo-inositol media on its own, with improvement in motility percentages (mainly rapid and MP speed groups) and kinematic parameters (increased NP velocity kinematics and decreased linear and vigor kinematics). The great effect on inducing hyperactivation and the increases observed in the AR of the HM subpopulation after HD-C exposure are possibly due to the presence of progesterone in the medium, and are in accordance with the literature [69,72,73]. 

Although HD-C did not affect MMP, a significant decrease in ROS percentages was evident when media was considered as a fixed factor. These observations could be a result of myo-inositol’s antioxidant interactions within the media. Subfertile spermatozoa produce large amounts of ROS, which impair various functional characteristics of spermatozoa and prevent sperm capacitation [69,72,73]. HD-C exposure counteracts these detrimental effects by improving sperm functionality, such as motility and hyperactivation, and reducing ROS. In addition, myo-inositol can probably improve the functionality of spermatozoa which may have altered CatSper channels and therefore do not respond to progesterone [62]. HD-C thus has the ability to interact with various functional sperm qualities and therefore augment the different fertilizing properties.

### 4.4. Dopamine

Dopamine is present in semen and oviduct fluids, and dopamine receptors have been reported in spermatozoa [13]. In accordance with Cariati and colleagues [74], we observed no significant effect of dopamine on sperm vitality percentages in subpopulations. However, dopamine improved the LM subpopulations’ vitality percentages to the extent that differences between the subpopulations were eliminated. Ramirez et al. [13] found that after a 3 h incubation in low levels of dopamine, the activation of DRD2-receptors in boar spermatozoa maintained sperm viability. As subpopulations were only incubated for 30 min in the current study, longer incubation time may have resulted in higher vitality percentages as compared to controls.

Our findings showed that dopamine improved motility percentages of both subpopulations and with media as a fixed factor. These findings are consistent with previous studies [29,74], which reported that dopamine could enhance sperm motility parameters following 30 min of incubation. Singh et al. [75] demonstrated that after being impaired by ethinyl estradiol, rats treated with dopamine had significantly improved sperm motility, MMP and ROS levels. 

In addition, higher dopamine concentrations increased sperm hyperactivation percentages (1 µM) in both subpopulations, as well as acrosome reaction percentages (100 nM and 1 µM), but only in the HM subpopulation. Our results are in agreement with Urra et al. [29], who reported increased sperm capacitation in vitro after dopamine stimulation, possibly due to increased protein–tyrosine phosphorylation and sperm motility. The apparent inability of dopamine to increase acrosome reaction in the LM subpopulations could be related to the fact that protein–tyrosine phosphorylation may be altered in LM subpopulations or subfertile males [76]. Interestingly, much higher dopamine concentrations (1 mM) have been reported to decrease both aspects; thus, dopamine may have a biphasic effect at varying concentrations [13,16,29]. Urra et al. [29] observed a decrease in acrosome integrity of stallion sperm after 6 hours’ incubation in high concentrations of dopamine (1 mM). Given that spermatozoa contain both β- and α-adrenergic receptors, it is plausible that high doses activate inhibitory β-adrenergic receptors, whereas low concentrations stimulate α-adrenergic receptors [77]. 

Our results confirmed that dopamine decreased ROS but improved MMP of the LM subpopulation. Since human infertility is associated with complex disturbances in hormones as well as ROS generation and disposal, exposure of sperm to dopamine may enhance its antioxidant and free radical scavenging properties [75].

### 4.5. Prolactin 

Prolactin receptors are located on the post-acrosomal region of the sperm head, neck, midpiece and principal piece of the sperm tail; however, the exact role of the hormone in sperm functionality is disputed in the literature [6]. With media considered as a fixed factor, prolactin exposure preserved sperm vitality percentages. Pujianto and colleagues [6] reported that prolactin may have the potential to inhibit sperm capacitation while simultaneously acting as a pro-survival factor that prevents spermatozoa from entering the default apoptotic pathway associated with ROS generation by mitochondria [78]. 

Serum prolactin levels are suggested to keep the male reproductive tract healthy, and prolactin is believed to improve sperm motility by increasing oxygen uptake by spermatozoa [79]. We observed that prolactin largely increased motility percentages in the LM subpopulations and when media was considered as a fixed factor. In contrast to the other media, prolactin did not appear to affect the kinematic parameters of the individual subpopulations. A possible explanation for prolactin mainly affecting the LM subpopulation could be that LM subpopulations, similar to asthenozoospermic males, may have compromised energy metabolism [80] as compared to HM subpopulations, and that prolactin may have a greater impact on improving mitochondrial ATP production in spermatozoa exhibiting low levels of motility. 

Prolactin has been suggested to affect a range of processes in vitro such as Ca^2+^ binding, maintaining mobility and attachment, as well as reducing capacitation time [81]. We observed that prolactin exposure significantly increased and maintained hyperactivation in both subpopulations. It is plausible that prolactin induces hyperactivation, as a dose-dependent increases of cAMP levels as well as utilization of fructose and glucose were demonstrated after exposure to bovine prolactin [82]. In contrast, no effect was found on subpopulation AR, as was also reported by Stovall and Shabanowitz [82] for similar prolactin concentrations (0–200 ng/mL) on sperm AR. The lack of effect on AR may reflect its suggested inhibitory effect on phosphotyrosine expression in spermatozoa [6]. While the HM subpopulation did not respond to prolactin exposure, the higher concentrations of prolactin (250 ng/mL and 500 ng/mL) did increase MMP values and reduce ROS in the LM subpopulation, which may reflect an increased energy metabolism in the LM subpopulations, as well as prolactin’s pro-survival effect [6,78].

## 5. Conclusions

This study highlighted the diverse functional characteristics of sperm subpopulations and recommends that their unique functional capabilities be considered when selecting an appropriate medium to enhance individual sperm attributes. As such, progesterone could be a potential medium to assist in inducing hyperactivation or improve progressivity in clinical scenarios, while HD-C has the potential to improve both HM and LM subpopulations’ sperm qualities. Dopamine and myo-inositol’s antioxidant capabilities could be utilized in semen to preserve MMP and vitality before processing/separation of semen samples, thereby increasing overall sperm motility while reducing ROS and subsequently DNA damage. Future investigations on possible pro-survival substances such as prolactin may allow for possible storage of spermatozoa under ART conditions, or, alternatively, be used as cultivation media for the processing of samples. Moreover, due to the heterogeneity of human ejaculates, it may be necessary to use media that contain a combination of biological substances in concentrations that are targeted for each subpopulation. Since the LM subpopulation closely mimics the functionality of sub-fertile semen samples, it is plausible that such media can be utilized in clinical scenarios to enhance the functionality of spermatozoa before ART treatment. Alternatively, exposing sub-fertile samples to a selection of the substances investigated could potentially increase the quality of retrieved spermatozoa so that more affordable ART techniques can be utilized; for example, intrauterine insemination (IUI) instead of intracytoplasmic sperm injection (ICSI). 

## Figures and Tables

**Figure 1 life-11-01250-f001:**
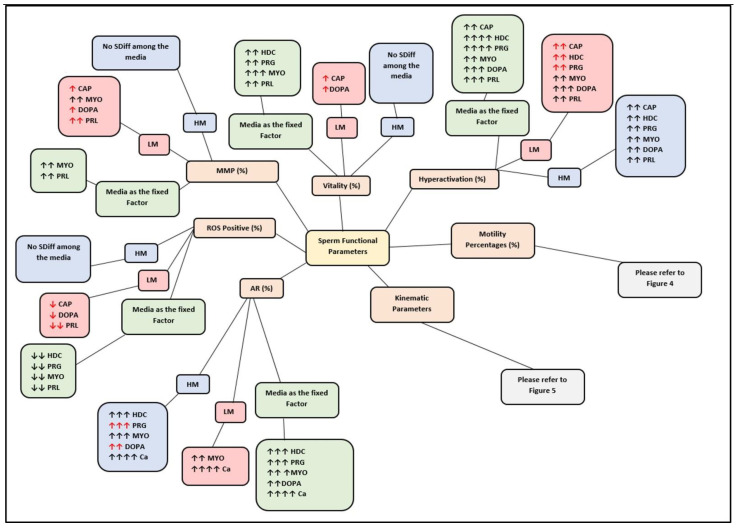
Bubble diagram displaying the overall effect of media on various functional parameters of spermatozoa for both subpopulations, and collectively with media as fixed factor. Red arrows (**↑** or **↓**) indicate a change in subpopulation (LM or HM) values resulting in elimination of significant differences between subpopulations. Two arrows (**↑↑** or ↓↓) indicate significant differences between the medium and HTF. Three arrows (**↑↑↑** or **↓↓↓**) indicate significant differences between the medium and both HTF and CAP. Four arrows (**↑↑↑↑** or **↓↓↓↓**) indicate significant differences between the medium and HTF, CAP and other media. Student’s *t*-test or the Mann–Whitney test (for non-parametric data) in addition to one-way ANOVA (for parametric distributions) or Kruskal–Wallis test (for non-parametric distributions) were used throughout unless indicated differently; *p* < 0.05. AR, acrosome reaction; CAP, capacitating-HTF; Ca-ionophore, calcium-ionophore; DOPA, dopamine; HD-C, HD capacitating medium; HM, high motile subpopulation; HTF, human tubal fluid; LM, low motile subpopulation; MMP, mitochondrial membrane potential; MYO, myo-inositol; PRG, progesterone; PRL, prolactin; ROS, positive reactive oxygen species; SDiff, significant difference.

**Figure 2 life-11-01250-f002:**
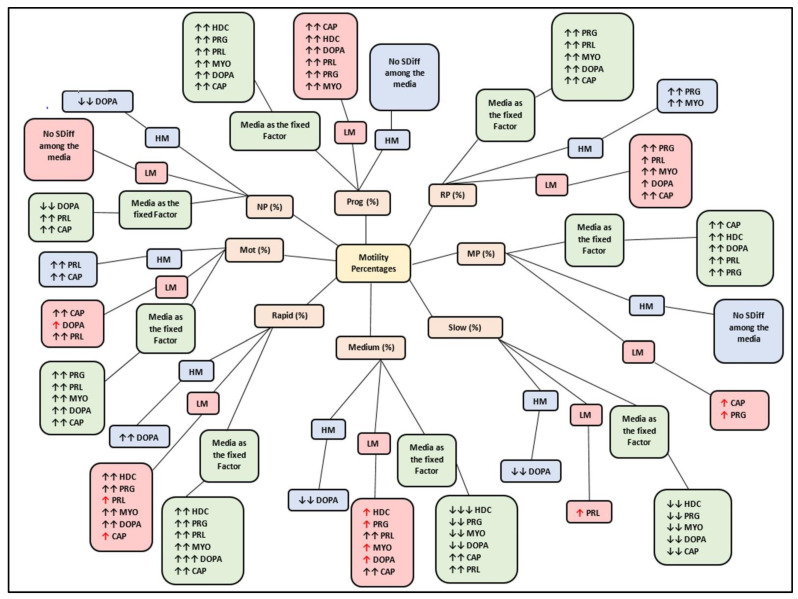
Bubble diagram displaying the overall effect of media on various motility percentages of spermatozoa for both subpopulations, and collectively with media as fixed factor. Red arrows (**↑** or **↓**) indicate a change in subpopulation (LM or HM) values resulting in elimination of significant differences between subpopulations. Two arrows (**↑↑** or **↓↓**) indicate a significant difference between the medium and HTF. Three arrows (**↑↑↑** or **↓↓↓**) indicate a significant difference between the medium and both HTF and CAP. Four arrows (**↑↑↑↑** or **↓↓↓↓**) indicate a significant difference between the medium and HTF, CAP and other media. CAP, capacitating-HTF; DOPA, dopamine; HD-C, HD capacitating medium; HM, high motile subpopulation; HTF, human tubal fluid; LM, low motile subpopulation; Mot, total motility; MP, medium progressive; MYO, myo-inositol; NP, non-progressive; PRG, progesterone; PRL, prolactin; Prog, progressive motility; RP, rapid progressive; SDiff, significant difference.

**Figure 3 life-11-01250-f003:**
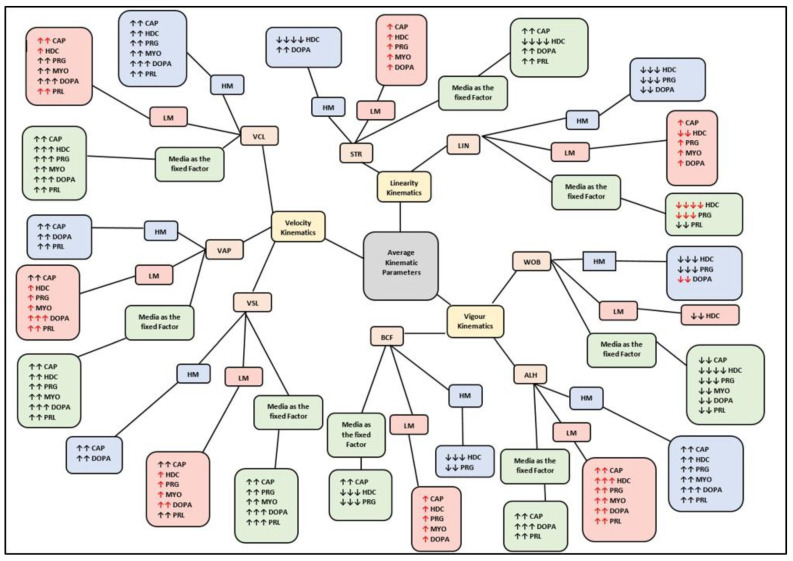
Bubble diagram displaying the overall effect of media on various motile/average kinematic parameters of spermatozoa for both subpopulations, and collectively with media as fixed factor. Red arrows (**↑** or **↓**) indicate a change in subpopulation (LM or HM) values resulting in elimination of significant differences between subpopulations. Two arrows (**↑↑** or **↓↓**) indicate a significant difference between the medium and HTF. Three arrows (**↑↑↑** or **↓↓↓**) indicate a significant difference between the medium and both HTF and CAP. Four arrows (**↑↑↑↑** or **↓↓↓↓**) indicate a significant difference between the medium and HTF, CAP and other media. ALH, amplitude of lateral head displacement; BCF, beat cross frequency; CAP, capacitating-HTF; DOPA, dopamine; HD-C, HD capacitating medium; HM, high motile subpopulation; HTF, human tubal fluid; LIN, linearity; LM, low motile subpopulation; MYO, myo-inositol; PRG, progesterone; PRL, prolactin; STR, straightness; VAP, average path velocity; VCL, curvilinear velocity; VSL, straight-line velocity; WOB, wobble.

**Figure 4 life-11-01250-f004:**
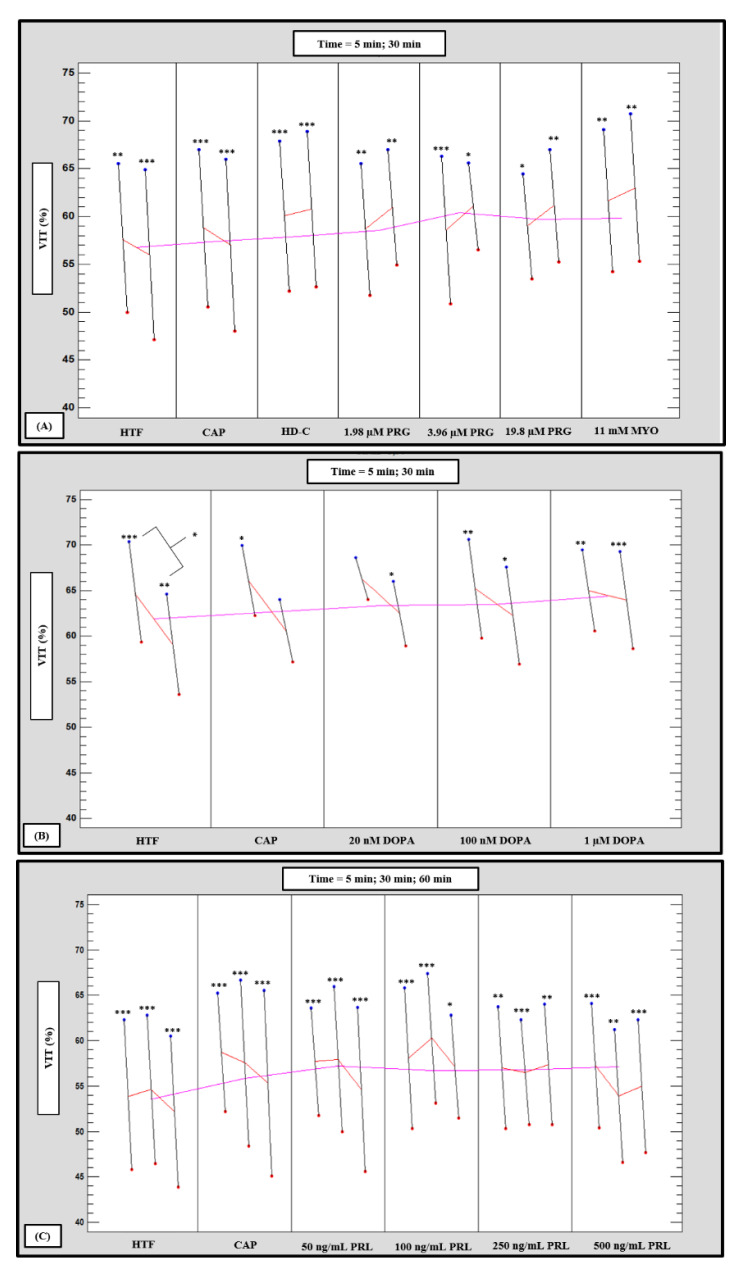
Multivariable chart comparing the mean percentage viable spermatozoa of high motile (HM, blue dots) and low motile (LM, red dots) sperm subpopulations at different time points after treatment with: (**A**) HTF, CAP, HD-C, PRG and MYO; (**B**) HTF, CAP and DOPA; and (**C**) HTF, CAP and PRL. Significant differences between subpopulations for individual media and time points are indicated as an asterisk (* *p* < 0.05, ** *p* < 0.01 and *** *p* < 0.001). CAP, capacitating-HTF; DOPA, dopamine; HD-C, HD capacitating medium; HM, high motile subpopulation; HTF, human tubal fluid; LM, low motile subpopulation; MYO, myo-inositol; PRG, progesterone; PRL, prolactin; VIT, vitality.

**Figure 5 life-11-01250-f005:**
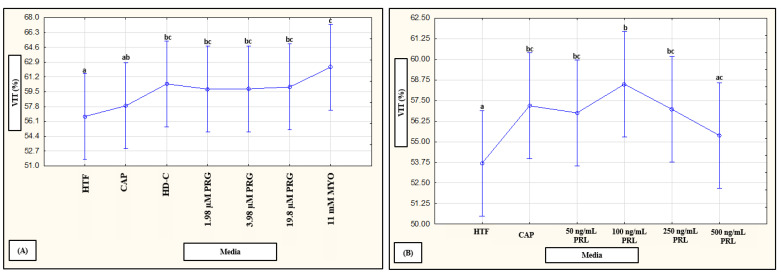
Pooled data of both high motile (HM) and low motile (LM) subpopulations and various time points (5, 30 min) as analyzed in a mixed model repeated measures ANOVA to determine significant (*p* < 0.01) interactions between media as fixed factor on percentage viable spermatozoa. Effect of (**A**) HTF, CAP, HD-C, PRG and MYO (F (6540) = 3.29, *p* < 0.01), and (**B**) HTF, CAP and PRL (F (5630) = 4.26, *p* < 0.01) on percentage viable spermatozoa. Vertical bars denote 0.95 confidence intervals and bars labelled with different superscript letters (a, b and c) were significantly different (*p* < 0.01). The Fisher LSD was used for the post hoc test and reports. CAP, capacitating-HTF; HD-C, HD capacitating medium; HTF, human tubal fluid; MYO, myo-inositol; PRL, prolactin; PRG, progesterone; VIT, vitality.

**Figure 6 life-11-01250-f006:**
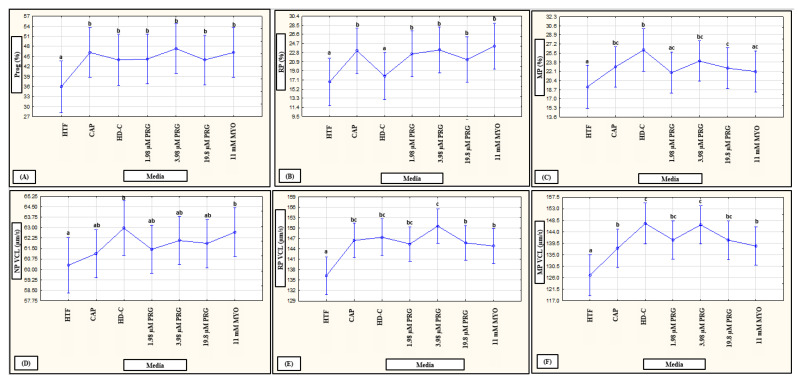
Pooled data of both high motile (HM) and low motile (LM) subpopulations and various time points (5, 30 min) as analyzed in a mixed model repeated measures ANOVA to determine the effect of media as a fixed factor. Effect of HTF, CAP, HD-C, PRG and MYO on: (**A**) progressivity (F (6527) = 3.54, *p* < 0.01); (**B**) rapid progressive (RP) speed groups (F (6528) = 2.92, *p* < 0.01) (**C**) medium progressive (MP) speed groups (F (6527) = 5.40, *p* < 0.01); (**D**) non-progressive (NP) VCL (F (6529) = 1.13, *p* = 0.34); (**E**) RP VCL (F (6524) = 7.15, *p* = 0.34); and (**F**) MP VCL (F (6520) = 6.16, *p* = 0.34). Vertical bars denote 0.95 confidence intervals and bars labelled with different superscript letters (a, b and c) were significantly different (*p* < 0.01). The Fisher LSD was used for the post hoc test and reports. CAP, capacitating-HTF; HD-C, HD capacitating medium; HTF, human tubal fluid; MP, medium progressive; MYO, myo-inositol; NP, non-progressive; PRG, progesterone; Prog, progressive; RP, rapid progressive; VCL, curvilinear velocity.

**Figure 7 life-11-01250-f007:**
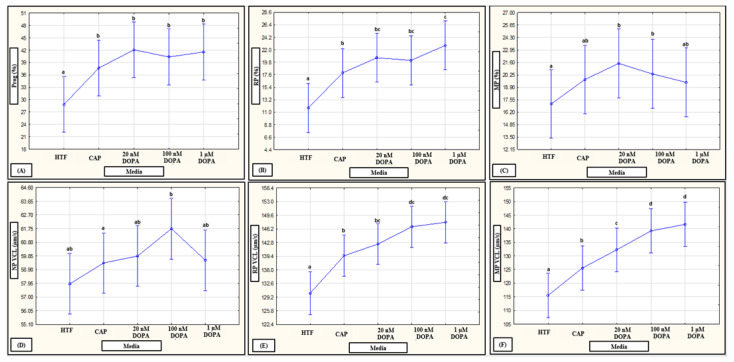
Pooled data of both high motile (HM) and low motile (LM) subpopulations and various time points (5, 30 min) as analyzed in a mixed model repeated measures ANOVA to determine the effect of media as a fixed factor. Effect of HTF, CAP and DOPA on: (**A**) progressivity (F (4452) = 11.41, *p* < 0.01); (**B**) rapid progressive (RP) speed groups (F (4452) = 14.67, *p* < 0.01); (**C**) medium progressive (MP) speed groups (F (4452) = 2.30, *p* = 0.06); (**D**) non-progressive (NP) VCL (F (4438) = 2.37, *p* = 0.05) (**E**) RP VCL (F (4438) = 23.82, *p* = 0.34); and (**F**) MP VCL (F (4421) = 15.24, *p* = 0.34). Vertical bars denote 0.95 confidence intervals and bars labelled with different superscript letters (a, b and c) were significantly different (*p* < 0.01). The Fisher LSD was used for the post hoc test and reports. CAP, capacitating-HTF; DOPA, dopamine; HTF, human tubal fluid; MP, medium progressive; NP, non-progressive; Prog, progressive; RP, rapid progressive; VCL, curvilinear velocity.

**Figure 8 life-11-01250-f008:**
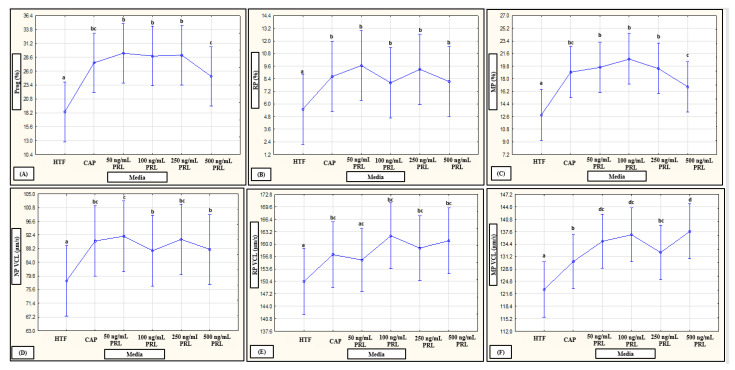
Pooled data of both high motile (HM) and low motile (LM) subpopulations and various time points (5, 30 min) as analyzed in a mixed model repeated measures ANOVA to determine the effect of media as a fixed factor. Effect of HTF, CAP and PRL on: (**A**) progressivity (F (5741) = 16.28, *p* < 0.01); (**B**) rapid progressive (RP) speed groups (F (5741) = 6.16, *p* < 0.01); (**C**) medium progressive (MP) speed groups (F (5741) = 13.25, *p* < 0.01); (**D**) non-progressive (NP) VCL (F (5730) = 12.61, *p* < 0.01); (**E**) RP VCL (F (5425) = 3.11, *p* < 0.01); and (**F**) MP VCL (F (5661) = 7.90, *p* < 0.01). Vertical bars denote 0.95 confidence intervals and bars labelled with different superscript letters (a, b and c) were significantly different (*p* < 0.01). The Fisher LSD was used for the post hoc test and reports. CAP, capacitating-HTF; HTF, human tubal fluid; MP, medium progressive; NP, non-progressive; Prog, progressive; PRL, prolactin; RP, rapid progressive; VCL, curvilinear velocity.

**Figure 9 life-11-01250-f009:**
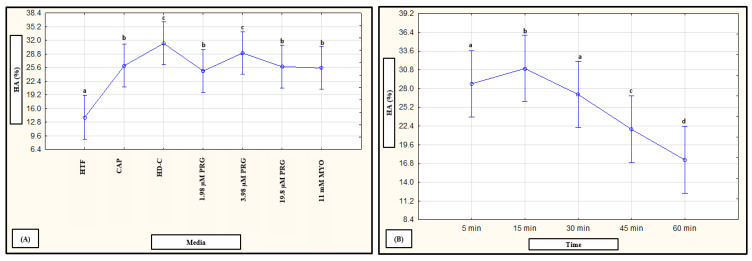
Mixed model repeated measures ANOVA to determine the effect media and time as individual fixed factors on percentage hyperactivation. (**A**) Effect of HTF, CAP, HD-C, PRG and MYO (F (6, 1252) = 33.06, *p* < 0.01). (**B**) Effect of time (5, 15, 30, 45 and 60 min) for pooled data of HTF, CAP, HD-C, PRG and MYO (F (4, 1250) = 60.35, *p* < 0.01). Vertical bars denote 0.95 confidence intervals and bars labelled with different letters were significantly different (*p* < 0.01). The Fisher LSD was used for the post hoc test and reports. CAP, capacitating-HTF; HD-C, HD capacitating medium; HA, hyperactivation; HTF, human tubal fluid; MYO, myo-inositol; PRG, progesterone.

**Figure 10 life-11-01250-f010:**
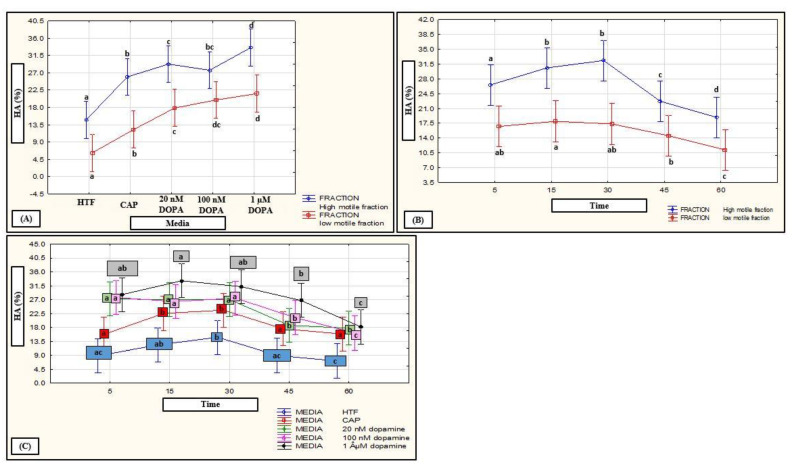
Mixed model repeated measures ANOVA to determine interactions between media and time as individual fixed factors and combined on percentage hyperactivation. (**A**) Interaction of media (HTF, CAP and DOPA) on HM and LM subpopulations (F (4, 931) = 2.45, *p* = 0.04). (**B**) Interaction of time (5, 15, 30, 45 and 60 min) on HM and LM subpopulations for pooled data of HTF, CAP and DOPA (F (4, 931) = 3.87, *p* < 0.01). (**C**) Interaction of media (HTF, CAP and DOPA) and time (5, 15, 30, 45 and 60 min) (F (16, 931) = 1.77, *p* = 0.03). Vertical bars denote 0.95 confidence intervals and bars labelled with different letters were significantly different (*p* < 0.01). The Fisher LSD was used for the post hoc test and reports. CAP, capacitating-HTF; DOPA; dopamine; HA, hyperactivation; HM, high motile; HTF, human tubal fluid; LM, low motile.

**Figure 11 life-11-01250-f011:**
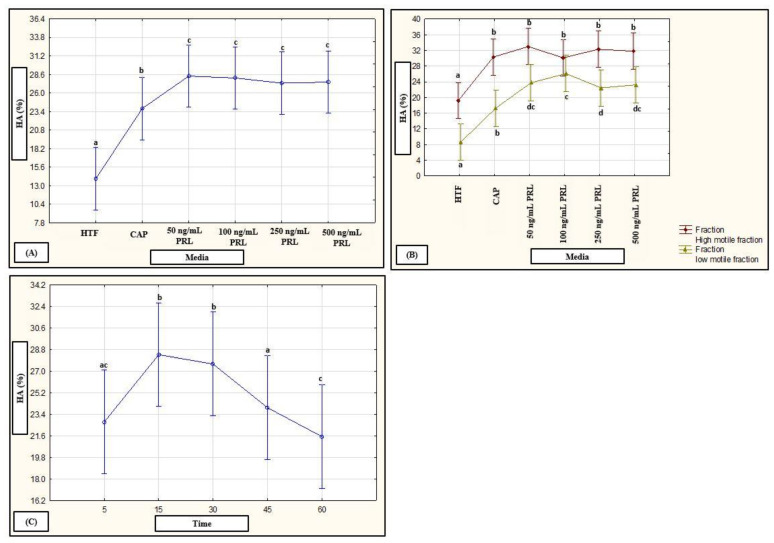
Mixed model repeated measures ANOVA to determine the effect of media and time as fixed factors as well as interactions between media and subpopulations on percentage hyperactivation. (**A**) Effect of HTF, CAP and PRL (F (5, 1357) = 52.15, *p* < 0.01). (**B**) Interaction between media (HTF, CAP and PRL) and subpopulations (F (5, 1357) = 3.66, *p* < 0.01). (**C**) Effect of time (5, 15, 30, 45 and 60 min) for pooled data of HTF, CAP and PRL (F (4, 1357) = 18.09, *p* < 0.01). Vertical bars denote 0.95 confidence intervals and bars labelled with different letters were significantly different (*p* < 0.01). The Fisher LSD was used for the post hoc test and reports. CAP, capacitating-HTF; HA, hyperactivation; HM, high motile; HTF, human tubal fluid; LM, low motile; PRL, prolactin.

**Figure 12 life-11-01250-f012:**
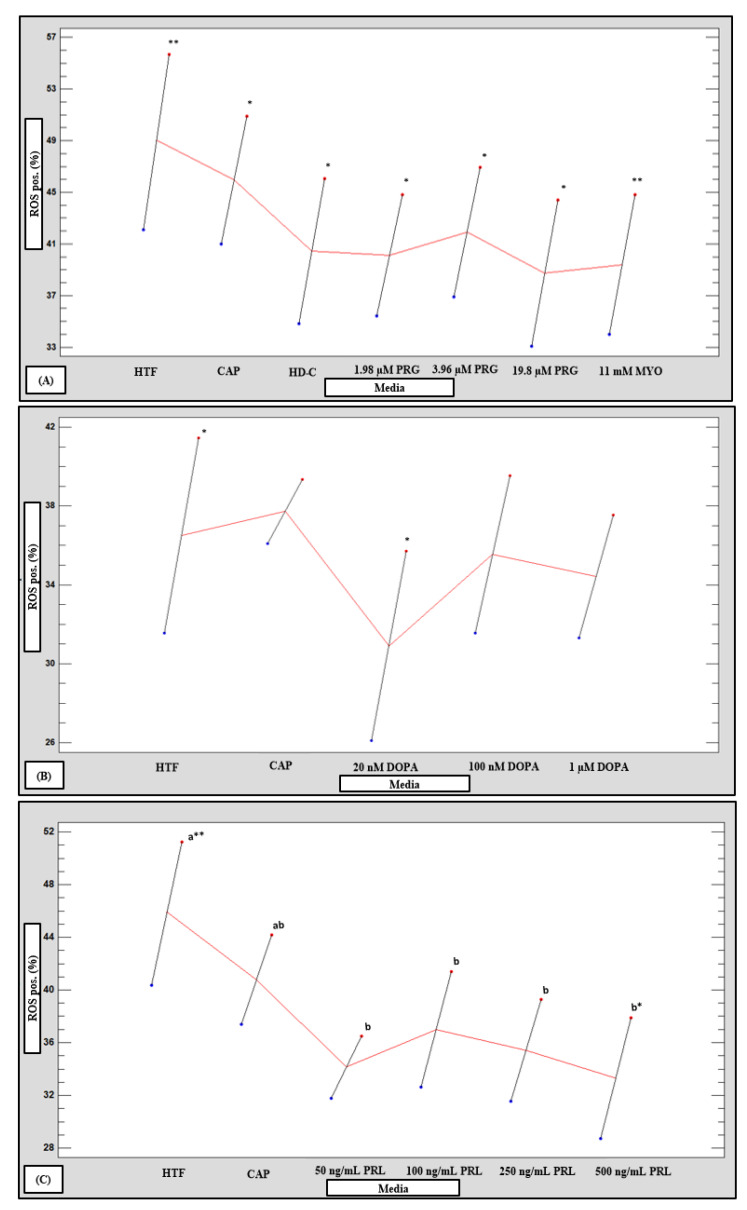
Multivariable chart comparing the mean percentage positive reactive oxygen species spermatozoa of high motile (HM, blue dots) and low motile (LM, red dots) sperm subpopulations after treatment with various media (*n* = 20). Comparison of HM and LM subpopulations after treatment with: (**A**) HTF, CAP, HD-C, PRG and MYO; (**B**) HTF, CAP and DOPA; and (**C**) HTF, CAP and PRL. Vertical bars labelled with different letters (a, b, c) were significantly different between media for individual subpopulations (*p* < 0.05). Vertical bars labelled with an asterisk were significantly different between subpopulations for individual media (* *p* < 0.05, ** *p* < 0.01). CAP, capacitating-HTF; DOPA, dopamine; HD-C, HD capacitation medium; HTF, human tubal fluid; HM, high motile subpopulation; LM, low motile subpopulation; MYO, myo-inositol; PRG, progesterone; PRL, prolactin; ROS pos., positive reactive oxygen species.

**Figure 13 life-11-01250-f013:**
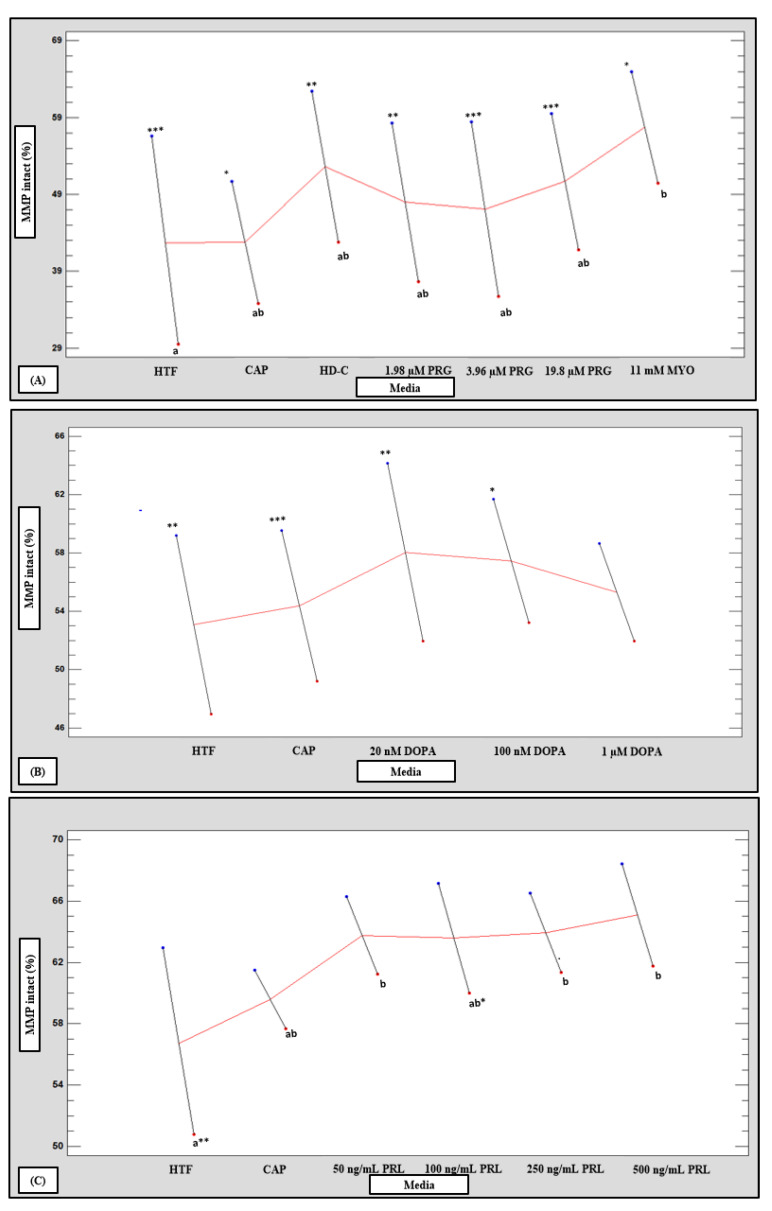
Multivariable chart comparing the mean percentage intact mitochondrial membrane potential (MMP) of the low (LM, red dots) and high (HM, blue dots) motile sperm subpopulation after incubation in various media (*n* = 20). Comparison of percentage intact MMP of LM and HM subpopulations after treatment with: (**A**) HTF, CAP, HD-C, PRG and MYO; (**B**) HTF, CAP and DOPA; and (**C**) HTF, CAP and PRL. Values labelled with different letters (a, b, c) were significantly different between the various media for individual subpopulations. Values with asterisks were significantly different between the HM and LM subpopulations for individual media (* *p* < 0.05, ** *p* < 0.01 and *** *p* < 0.001). CAP, capacitating-HTF; DOPA, dopamine; HD-C, HD capacitation medium; HTF, human tubal fluid; HM, high motile subpopulation; LM, low motile subpopulation; MMP, mitochondrial membrane potential; MYO, myo-inositol; PRG, progesterone; PRL, prolactin.

**Figure 14 life-11-01250-f014:**
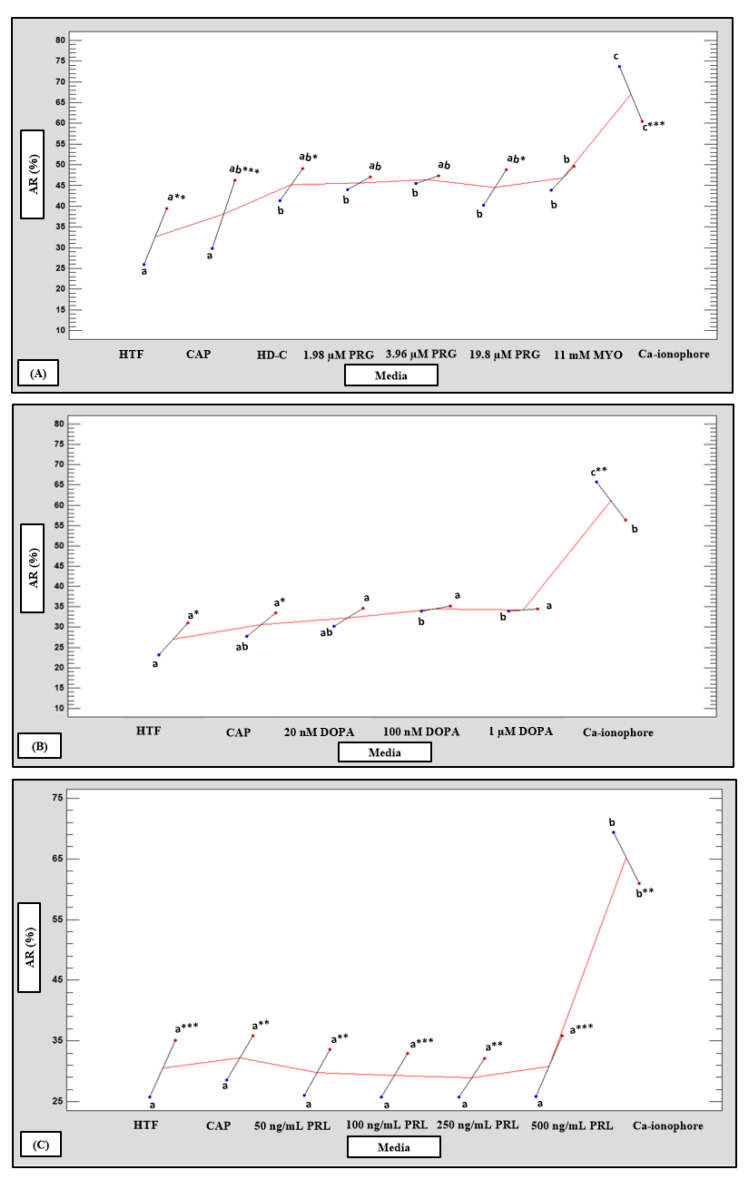
Multivariable chart displaying the percentage acrosome reaction (AR) of the low (LM, **red** dots) and high (HM, blue dots) motile sperm subpopulation after incubation in various media (*n* = 20). Comparison of percentage AR for the LM and HM motile sperm subpopulation after incubation in: (**A**) HTF, CAP, HD-C, PRG, MYO and Ca-ionophore; (**B**) HTF, CAP, DOPA and Ca-ionophore; and (**C**) HTF, CAP, PRL and Ca-ionophore. Values labelled with different letters (a, b, c) were significantly different between the various media for individual subpopulations. Values with asterisks were significantly different between the HM and LM subpopulations for individual media (* *p* < 0.05, ** *p* < 0.01 and *** *p* < 0.001). AR, acrosome reaction; Ca-ionophore, calcium-ionophore; CAP, capacitating-HTF; DOPA, dopamine; HD-C, HD capacitation media; HTF, human tubal fluid; HM, high motile subpopulation; LM, low motile subpopulation; MMP, mitochondrial membrane potential; MYO, myo-inositol; PRG, progesterone; PRL, prolactin.

**Table 1 life-11-01250-t001:** Basic semen parameters (mean ± SD) of donor samples used in this investigation (*n* = 160).

	Mean ± SD	95% C.I
Total Mot (%)	54.5 ± 18.5	51.7–57.3
Prog Mot (%)	27.3 ± 16.5	24.8–29.8
MPT (10^6^/ejaculate)	28.3 ± 27.0	24.2–32.4
pH	7.4 ± 0.2	7.4–7.5
Viscosity (cP)	10.8 ± 11.1	9.1–12.5
Volume (mL)	3.2 ± 2.9	2.8–3.7
Conc (10^6^/mL)	58.6 ± 42.8	52.2–65.1
Conc (10^6^/ejaculate)	161.3 ± 143.5	139.6–183.0
Vitality (%)	69.9 ± 10.7	68.3–71.6
Normal (%)	5.7 ± 4.6	5.0–6.4

C.I, confidence interval; Conc, concentration; cP, centipoise; MPT, mucous penetration test; Prog Mot, progressive motility; SD, standard deviation; Total Mot, total motility.

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
