# Peer review of "Progesterone, Myo-Inositol, Dopamine and Prolactin Present in Follicular Fluid Have Differential Effects on Sperm Motility Subpopulations"

_life, 2021, doi:10.3390/life11111250_

Round 1

Reviewer 1 Report

The article entitled “Progesterone, Myo-inositol, Dopamine and Prolactin Present in Follicular Fluid Have Differential Effects on Sperm Motility  Subpopulations” aims to investigate the response of two sperm motility subpopulations (mimicking the functionality of potentially fertile and sub- fertile semen samples) to different biological substances present in the female reproductive tract such as Progesterone, Myo-inositol, Dopamine and Prolactin Present. The manuscript is well conceived, introduction is clear and adequate, results are well described and the use of bubble diagram aid to better understand the complexity of the experimental design and related results.

My only major concern is about the explanation of the experimental design in particular, for this study have been enrolled a total 160 patients but is unclear the number of patients who are used for each experimental group, and in addition how many are the replicates for each experimental group.

Author Response

Response: Noted and additional information was added accordingly.

In total, 44 donors (not patients) were used, from which 160 semen samples were obtained over the period of the investigation. As human semen is extremely heterogeneous, semen parameters varies between each ejaculate (even from the same male), thus no ejaculate will have the same quality/results as the one proceeding it. Multiple samples were therefore collected from each donor, provided there was at least 3 days abstinence between sample collections.

Since sperm are extremely sensitive to both temperature and time, we divided the assessment of the media into groups to reduce the negative effect of time on analysis of the various functional parameters. The effects of progesterone, myo-inositol and HD-C were investigated as a group (Group 1) and thereafter separate groups involving dopamine concentrations (Group 2) and prolactin concentrations (Group 3) were investigated. As part of each group, HTF and capacitating HTF (CAP) were used as negative and positive controls, respectively. Furthermore, due to testing several concentrations of most of the biological substances, we could not conduct all the experimental procedures on all the functional parameters at once, thus parameters were grouped and analysed using different semen samples on different days. The vitality assessments for Group 1 and Group 2 were done using 20 semen samples each (20 x 2 = 40 semen samples). Motility and hyperactivation were analysed together, using 20 semen samples each for Group 1 and Group 2 (20 x 2 = 40 semen samples). Vitality, motility and hyperactivation were assessed together using the same semen samples for dopamine due to less concentrations being investigated (20 semen samples). MMP, ROS and AR were assessed together on the same semen samples for each media group (20 x 3 = 60 semen samples). We therefore used a total number of 160 semen samples during this study. 

Line 118 – 121: “One hundred and sixty human semen samples from forty-four donors were obtained via masturbation as part of a donor program (Division of Medical Physiology, Department of Biomedical Sciences, Stellenbosch University and Department of Medical Bioscience, University of the Western Cape) after two to three days of sexual abstinence.”

Line 155 – 165: “Due to the negative effect that time has on sperm functionality, selected media was grouped together and groups were assessed on separate samples and occasions; however, controls were included throughout. To this effect, HD-C, myo-inositol and progesterone were grouped and analyzed together, where after dopamine concentrations and prolactin concentrations were respectively grouped and analyzed together. Furthermore, due to time constraints, the investigation of functional parameters was additionally grouped and analyzed on different samples and occasions as follows: vitality was assessed on 20 samples; motility and hyperactivation were assessed together on 20 samples; and MMP, ROS and AR were assessed together on 20 samples. For the dopamine group, vitality, motility and hyperactivation could be analyzed together on the same 20 samples due to less concentrations being investigated.”  

Reviewer 2 Report

The study appears to bring new, interesting, and complex data concerning the role of progesterone (P4), myo-inositol, dopamine (DOPA) and prolactin (PRL) on sperm population separated regarding the motility. Overall, the results are interesting and, in my opinion, valuable, nevertheless, after careful insight I have some comments and doubts.

The introduction is well constructed and written, however, the question arises about newer data for ART success rate? The clear hypothesis of the study is not specified enough.

If the LM and HM as imitating models of sub-fertile and fertile semen samples were described in other studies,  the references should appear in the text.

Because the Authors prepared the P4 and PRL in DMSO and HCL, respectively, did the effect of mentioned chemicals alone was checked on the sperm parameters, especially on sperm vitality.  Did positive and /or negative controls were performed during the experiments described in the manuscript?

Why did the Authors decide to extend the time of sperm incubation only with PRL (to 60 min) during the sperm vitality test?

In the case of sperm vitality, ROS, MMP, and acrosome reaction, there it would be a huge advance if the Authors presented the Figures/images regarding these analyses in the Results section.

Figure 2 – there is (B) missing from this Figure.

Why the results concerning the effect of P4 and MYO are presented together, while data of DOPA effect separately. The reason should be clearly highlighted in the M&M section.

Why the results of the effect of DOPA on sperm vitality are not present in the results section (Fig.3), however, the data are in the bubble diagram (Fig.1).

Why in supplementary file legends did the Authors describe n=20. In my opinion, it is a mistake.

The Authors mentioned that future investigations are needed to explain the role of PRL, DOPA, P4, myoinositol, and medium, I wonder if the Authors tried to test the influence of co-treatments/interactions tested substances on sperm motility subpopulations.

The abbreviation of names that appeared for the first time in the text of a manuscript should be explained, for example, DGC. Please check others.

There are a few typos and punctuation errors.

Author Response

Comments:

The study appears to bring new, interesting, and complex data concerning the role of progesterone (P4), myo-inositol, dopamine (DOPA) and prolactin (PRL) on sperm population separated regarding the motility. Overall, the results are interesting and, in my opinion, valuable, nevertheless, after careful insight I have some comments and doubts.

  • The introduction is well constructed and written, however, the question arises about newer data for ART success rate? The clear hypothesis of the study is not specified enough.

Response: Noted and alterations were made accordingly. More recent data for ART success rate was included (see Line 37-38) and the last paragraph of the Introduction was rewritten (see Line 90-102) to include a clear hypothesis, as recommended.

  • If the LM and HM as imitating models of sub-fertile and fertile semen samples were described in other studies, the references should appear in the text.

Response: Noted and edited accordingly by adding the necessary references.

  • Because the Authors prepared the P4 and PRL in DMSO and HCL, respectively, did the effect of mentioned chemicals alone was checked on the sperm parameters, especially on sperm vitality.  Did positive and /or negative controls were performed during the experiments described in the manuscript?

Response: P4 and PRL stock solutions were prepared using DMSO and HCL according to suppliers’ instructions or to protocols described in previous studies, as indicated in the Materials and Methods section. Working solutions of P4 and PRL were prepared in CAP medium and osmolality and pH of the working solutions were adjusted accordingly. The possible effect of DMSO and HCL alone was not tested separately on any of the functional sperm parameters, since the volume/concentration of these chemicals were extremely low in the media used to test the final concentrations of biological substances (e.g. diluted from the stock solution: HCl, 20x – 200x; DMSO, 1000x – 10 000x). The negative and positive controls were HTF and CAP medium respectively (see Section 2.3) and were used for comparisons of all media and functional parameters. 

  • Why did the Authors decide to extend the time of sperm incubation only with PRL (to 60 mn) during the sperm vitality test?

Response: Initially, we did not see clear results after 30 minutes of incubation with PRL concentrations, as was observed for progesterone, myo-inositol and dopamine. Since previous studies reported results with PRL after longer incubation periods, we decided to extend the time points for PRL in our study. Due to the fact that PRL was the last biological substance to be investigated, we could not add any additional time points to the other three substances.  

  • In the case of sperm vitality, ROS, MMP, and acrosome reaction, there it would be a huge advance if the Authors presented the Figures/images regarding these analyses in the Results section.

Response: The results for sperm vitality, ROS, MMP and acrosome reaction are presented in Figures 2, 12, 13 and 14. Since the study focussed on quantitative data generated by using well-known techniques, images illustrating qualitative results are not included, but rather only quantitative data as various graphs in figures.

  • Figure 2 – there is (B) missing from this Figure.

Response: Noted and corrected accordingly.

  • Why the results concerning the effect of P4 and MYO are presented together, while data of DOPA effect separately. The reason should be clearly highlighted in the M&M section.

Response: Noted and additional information was added accordingly.

Line 155 – 165: “Due to the negative effect that time has on sperm functionality, selected media was grouped together and groups were assessed on separate samples and occasions; however, controls were included throughout. To this effect, HD-C, myo-inositol and progesterone were grouped and analyzed together, where after dopamine concentrations and prolactin concentrations were respectively grouped and analyzed together. Furthermore, due to time constraints, the investigation of functional parameters was additionally grouped and analyzed on different samples and occasions as follows: vitality was assessed on 20 samples; motility and hyperactivation were assessed together on 20 samples; and MMP, ROS and AR were assessed together on 20 samples. For the dopamine group, vitality, motility and hyperactivation could be analyzed together on the same 20 samples due to less concentrations being investigated.”  

  • Why the results of the effect of DOPA on sperm vitality are not present in the results section (Fig.3), however, the data are in the bubble diagram (Fig.1).

Response: The effect of DOPA on sperm vitality is illustrated in Figure 2B, which indicates both the effect of time and differences between the HM and LM subpopulations (also summarized in red bubble in Figure 1). Figure 3, however, illustrates the significant interactions of media as fixed factor, for which only progesterone, myo-inositol, HD-C (Fig. 3A) and prolactin (Fig. 3B) revealed significant results. Similarly, in Figure 1, in the green bubble for media as fixed factor, only significant interactions are displayed, thus DOPA is omitted in this case since no significant interactions were found among the controls and different concentrations of DOPA for both subpopulations.

  • Why in supplementary file legends did the Authors describe n=20. In my opinion, it is a mistake.

Response: Noted and additional information was added accordingly.

In total, 44 donors (not patients) were used, from which 160 semen samples were obtained over the period of the investigation. As human semen is extremely heterogeneous, semen parameters varies between each ejaculate, thus no ejaculate will have the same quality/results as the one proceeding it. Multiple samples were therefore collected from each donor, provided there was at least 3 days abstinence between sample collections.

Since sperm are extremely sensitive to both temperature and time, we divided the assessment of the media into groups to reduce the negative effect of time on analysis of the various functional parameters. The effects of progesterone, myo-inositol and HD-C were investigated as a group (Group 1) and thereafter separate groups involving dopamine concentrations (Group 2) and prolactin concentrations (Group 3) were investigated. As part of each group, HTF and capacitating HTF (CAP) were used as negative and positive controls, respectively. Furthermore, due to testing several concentrations of most of the biological substances, we could not conduct all the experimental procedures on all the functional parameters at once, thus parameters were grouped and analysed using different semen samples on different days. The vitality assessments for Group 1 and Group 2 were done using 20 semen samples each (20 x 2 = 40 semen samples). Motility and hyperactivation were analysed together, using 20 semen samples each for Group 1 and Group 2 (20 x 2 = 40 semen samples). Vitality, motility and hyperactivation were assessed together using the same semen samples for dopamine due to less concentrations being investigated (20 semen samples). MMP, ROS and AR were assessed together on the same semen samples for each media group (20 x 3 = 60 semen samples). We therefore used a total number of 160 semen samples during this study. 

Line 118 – 121: “One hundred and sixty human semen samples from forty-four donors were obtained via masturbation as part of a donor program (Division of Medical Physiology, Department of Biomedical Sciences, Stellenbosch University and Department of Medical Bioscience, University of the Western Cape) after two to three days of sexual abstinence.”

Line 155 – 165: “Due to the negative effect that time has on sperm functionality, selected media was grouped together and groups were assessed on separate samples and occasions; however, controls were included throughout. To this effect, HD-C, myo-inositol and progesterone were grouped and analyzed together, where after dopamine concentrations and prolactin concentrations were respectively grouped and analyzed together. Furthermore, due to time constraints, the investigation of functional parameters was additionally grouped and analyzed on different samples and occasions as follows: vitality was assessed on 20 samples; motility and hyperactivation was assessed together on 20 samples; and MMP, ROS and AR were assessed together on 20 samples. For the dopamine group, vitality, motility and hyperactivation could be analyzed together on the same 20 samples due to less concentrations being analyzed.”  

  • The Authors mentioned that future investigations are needed to explain the role of PRL, DOPA, P4, myoinositol, and medium, I wonder if the Authors tried to test the influence of co-treatments/interactions tested substances on sperm motility subpopulations.

Response: We are in the process of investigating media in which the four biological substances are combined, but we do not as yet have concrete data on such interactions and its effect on the two subpopulations. It was thus also mentioned in the Conclusion that such combinations should be tested.

  • The abbreviation of names that appeared for the first time in the text of a manuscript should be explained, for example, DGC. Please check others.

Response: Note and changed accordingly.

  • There are a few typos and punctuation errors.

Response: Noted and corrected accordingly.